# Functional classes of SNPs related to psychiatric disorders and behavioral traits contrast with those related to neurological disorders

**Mark A. Reimers** [1]*, **Kenneth S. Kendler** [2]*

**1** Institute for Quantitative Health Sciences and Engineering, Michigan State University, East Lansing, MI, United States of America, **2** Virginia Institute of Psychiatric and Behavioral Genetics, and Department of Psychiatry, Medical College of Virginia/Virginia Commonwealth University, Richmond, VA, United States of America

* reimersm@msu.edu (MAR); kenneth.kendler@vcuhealth.org (KSK)

**Data Availability Statement:** Raw data are available from the sources cited in Table 2. Processed data are available as S1 Table.

## Abstract

We investigated the functional classes of genomic regions containing SNPS contributing most to the SNP-heritability of important psychiatric and neurological disorders and behavioral traits, as determined from recent genome-wide association studies. We employed linkage-disequilibrium score regression with several brain-specific genomic annotations not previously utilized. The classes of genomic annotations conferring substantial SNP-heritability for the psychiatric disorders and behavioral traits differed systematically from the classes associated with neurological disorders, and both differed from the classes enriched for height, a biometric trait used here as a control outgroup. The SNPs implicated in these psychiatric disorders and behavioral traits were highly enriched in CTCF binding sites, in conserved regions likely to be enhancers, and in brain-specific promoters, regulatory sites likely to affect responses to experience. The SNPs relevant for neurological disorders were highly enriched in constitutive coding regions and splice regulatory sites.

## 1. Introduction

Recent studies (e.g. [1]) have found that little of the SNP-heritability for psychiatric disorders lies in coding regions. These results provoke the question: what kinds of genomic elements are relevant to each psychiatric disorder–which we term the 'functional genetic architecture' of the disorder–and do the functional genetic architectures of psychiatric disorders differ systematically from those of complex neurological disorders or behavioral or anthropometric traits? By comparing the functional genetic architectures of psychiatric disorders to those of neurological disorders, behavioral traits, and brain-related disorders, we sought to determine if the mechanisms of disorders differ systematically and how the resulting typology of illness relates to typology based on familial factors and/or SNP-based polygenic risk scores.

Twin and family studies have investigated the degree to which different psychiatric disorders share familial liability [2, 3]. With the development of polygenic risk scores (PRS),

**Funding:** KSK received a grant from the John Templeton Foundation (https://www.templeton.org). The title was "The Nature of Molecular Risk Variants for Psychiatric and Psychological Phenotypes") The funders had no role in study design, data collection and analysis, decision to publish, or preparation of the manuscript.

**Competing interests:** The authors have declared that no competing interests exist.

evidence for substantial genetic correlations across various psychiatric disorders was replicated and extended [2, 3] while the correlations across psychiatric and neurological disorders were limited [4]. These results are of interest outside the specialized area of psychiatric genetics because the familial/genetic relationships between psychiatric disorders are used as one primary method for clarifying nosologic boundaries between disorders [5].

However, a complementary approach to the genetic architecture of psychiatric and neurologic disorders examines the relative contributions of different functional classes of genomic elements, such as dynamic regulators, affecting response to experience, or constitutive regulators that may affect cell-type identity, coding regions etc. This is the approach taken here.

A separate important issue is whether the findings of psychiatric genetics can be integrated with the well-established findings of the life-history risk factors for mental illness [6, 7]. Although psychiatric GWAS implicate many brain-related genes, especially synaptic genes, it remains unclear how the genetic risk factors may be related to the well-documented environmental risk factors for illness. A simple hypothesis is the that the genetic risk factors for psychiatric disorders lie predominantly in DNA that dynamically regulates genes in response to changing environmental circumstances or bodily signals, rather than in DNA that determines protein products or cell-type identity.

Several groups have attempted to partition the common variant (SNP) heritability of select psychiatric disorders among different functional categories. Schork et al. [8] compared genetic contributions of different parts of coding genes and found that the untranslated regions accounted for more heritability than coding regions for schizophrenia; however, the authors noted that, because of the high linkage disequilibrium (LD) in the human genome, it is difficult to assign unambiguously a particular association signal to a particular SNP, and thereby to determine in which categories most heritability lies. This assignment is especially challenging for functional classes that are frequently juxtaposed on the genome, (e.g. transcription start sites (TSS) and promoters) so that SNPs in LD with a SNP in one functional class are often in high LD with a SNP in another class. Schork et al. [8] attempted to resolve this ambiguity by adding together all the annotated SNPs in LD with each SNP of genome wide significance, weighted by the LD $r^2$.

Finucane et al. [9] addressed the issue of LD more systematically using partitioned linkage disequilibrium score regression (LDSR). This method exploits the wide distribution of risk SNPs with small effects and is based on the idea that SNPs in high LD with classes of SNPs most relevant to risk will have systematically elevated chi-square association scores. Their initial presentation used a large set of diverse annotations from different sources, including some regulatory types; they offered a distribution of SNP-heritability among annotation classes for several disorders and traits. However, most of the annotations they used were not brain-specific, and significant improvements in the annotation of regulatory functions have been made since their use of generic ENCODE data. This is an opportune time to revisit the LDSR approach using more recent and brain-specific data.

The goals of this study are to characterize the functional genetic architecture of a range of psychiatric and neurological disorders and behavioral traits. We predicted that a preponderance of the heritability for psychiatric disorders and behavioral traits would be in regulatory sites, specifically enhancers, while most of the heritability for neurological disorders would be in protein coding regions. We further expected long non-coding RNAs (lncRNAs) to contribute to psychiatric disorders because they were highly expressed in developing human cortex [10]. However our LDSR-based method lacked statistical power to reliably estimate their enrichment, as detailed below.

## 2. Methods

### 2.1 Sources of data

We annotated 9.5M SNPs in the human genome (HG19) as follows. We downloaded from the LDSR github site certain key generic (i.e. tissue-independent) annotations (e.g. coding regions) used in [9]. We added selected several non-coding generic annotations from ENSEMBL, conservation data from UCSC and we included some brain-specific regulatory annotations based on chromatin data from RoadMap Epigenomics [11] and from PsychEN-CODE [12, 13]. One of these categories was brain-specific promoters (brain promoters from RoadMap, excluding the generic promoters annotated in ENSEMBL). These annotations and their sources are summarized in Table 1.

We generated several new kinds of annotations. One important source of regulatory variability is splicing. Splice regulatory are commonly found on either side of the splice junction, but they are poorly known or annotated. We assigned SNPs provisionally to these categories if they were located on introns within 70bp of an annotated splice junction and conserved across mammals.

Since we expected much of the heritability of psychiatric disorders to be in regulatory regions such as enhancers, we gathered and used annotations of enhancers from several sources, based on chromatin assays. However, although annotated enhancers (based on H3K27ac or ATAC chromatin peaks) from these studies showed significant enrichment among SNPs implicated by psychiatric and behavioral GWAS, none explained more than 20% of SNP-heritability of any phenotype using the LDSR model. Some reasons for this, and prospects for better enhancer annotations, are discussed below.

Therefore we decided to adopt the following strategy to identify probable enhancers. Our annotation classes included all the known specific non-coding elements of the genome, many of which are highly conserved. We reasoned that most of the remaining non-coding regions highly conserved across mammals (PhastCons > 0.5) were likely to be enhancers, even though not all would be active in the brain. One well-known problem with using conserved regions to identify enhancers is that enhancers are typically not well conserved across different orders of animals; furthermore there has likely been recent rapid evolution of regulatory sites affecting the human brain. This problem was partially addressed by using primate conservation data

**Table 1. Sources of genome annotations used in this study.**

| Annotation | Source | Reference | Comment | Proportion of SNPs |
|---|---|---|---|---|
| Promoter UCSC | LDSR | Finucane | | 0.0463 |
| TSS | LDSR | Finucane | | 0.0178 |
| Protein coding | LDSR | Finucane | | 0.0143 |
| 3' UTR | LDSR | Finucane | | 0.0036 |
| 5' UTR | LDSR | Finucane | | 0.0055 |
| Splice donor | Constructed | | 70 nt from start of intron and conserved | 0.0024 |
| Splice acceptor | Constructed | | 70 nt from end of intron and conserved | 0.0019 |
| Brain Promoter | RoadMap | RoadMap Epigenomics | | 0.0031 |
| Mammal Conserved | UCSC | | excluding other annotations | 0.0059 |
| Primate Conserved | UCSC | | excluding other annotations | 0.0136 |
| CTCF binding | PsychENCODE | | | 0.0194 |
| lncRNA | ENSEMBL | | | 5.00E-04 |
| micro-RNA | ENSEMBL | | | 6.40E-05 |
| ribosomal RNA | ENSEMBL | | | 8.90E-06 |

from UCSC; only 20% of these primate-conserved regions overlapped other mammal-conserved regions, consistent with the rapid evolution of brain enhancers in the primate lineage. This also gave us information about distinctively primate regulation, which we expected a priori would play a large role in human psychiatric disorders and behavior traits.

## 2.2 Class-specific heritability estimates

We used the LDSR procedure software LDSC provided by the Broad Institute (https://github.com/bulik/ldsc), and made the following modifications, both in line with their recommendations. First, two regions of very high linkage disequilibrium were excluded: the MHC region and the GPHN yin-yang region since both have strong associations with some psychiatric disorders and their leverage points would distort the regression. Second, the LDSR regression model tacitly assumes that all effect sizes within a category are comparable. However, the actual distribution of effect sizes observed in GWAS is very strongly right skewed and outliers can substantially distort least squares fits, such as those used in LDSR. We therefore winsorised the summary P values at $10^{-7}$, corresponding to a chi-square of 22.

Besides the categories reported here we also used several annotations of non-coding RNAs (long non-coding RNAs, microRNAs, and ribosomal RNAs). The proportions of SNPs with each of these annotations were less than 1 in 10,000, and the standard errors of heritability estimates for those classes from LDSR were almost all larger than the estimates. Thus, these SNP classes are omitted from this presentation.

Genome build made little difference to the results. Running LDSR for partitioned heritability on the same GWAS summaries using LD from either HG19 or HG38 had minimal impact on the heritability estimates. Since most of the GWAS results used here were reported initially in HG19, we used LDSR on this older build.

We obtained GWAS data from 22 studies of 21 brain-related phenotypes as listed in Table 2. We attempted to sample broadly from psychiatric disorders and behavioral traits [14–25], as well a selection of neurological or other brain-related disorders [26–32]. We tried to include GWAS from all disorders discussed in [4]; however the public data for several of these disorders were insufficiently dense to obtain plausible estimates via LDSC; we note that the list from [4] includes several brain-related disorders (e.g. stroke, AMD, chronic pain) that are not thought to be primary disorders of the central nervous system. We included GWAS from two well-studied biometric traits, height and BMI, as controls. There is some concern that estimates, from complex statistical methods, such as LDSR, may depend sensitively on the data set. We therefore include two independent GWAS of Alzheimer's disease to give a sense of how sensitively our estimates may depend on sample selected.

The LDSC program was downloaded in March 2019 and run using recommended settings. The LDSR estimates are unbiased, thus the LDSR method yields some negative heritability estimates when the standard error of the estimates exceeds the (positive) true $h^2$. The proportion of negative estimates of proportions of $h^2$ was consistent with what would be expected if one third of the categories contributed much lower SNP-heritability than the standard errors of the estimates. These negative estimates occurred mostly for those traits, which themselves have low SNP-heritability (mostly behavioral traits). Furthermore LDSR estimates for some categories had standard errors within a factor of two of the estimates themselves.

To reduce the error in estimates of $h^2$, we used an empirical Bayes (eBayes) approach. We started by observing that for annotation classes with well estimated heritabilities, (i.e. small standard errors), the estimates followed an approximately exponential distribution across different phenotypes. Therefore, we modeled the distribution of $h^2$ for each class across phenotypes by an exponential for all annotation classes. We estimated the parameter for each class

**Table 2. Sources of GWAS data used in this study.**

| Trait | Abbreviation | Study | Total N |
|---|---|---|---|
| Age-related Macular Degeneration | AMD | Fritsche et al. 2013 | 77255 |
| Alcohol Use Disorder | AUD | Walters et al. 2018 | 46568 |
| Alzheimer's disease | AD1 | Jansen et al. Nat Genet. 2019 | 455258 |
| Alzheimer's disease | AD2 | Kunkle et al. Nat Genet. 2019 | 63926 |
| Attention Deficit Hyperactivity Disorder | ADHD | Demontis et al. Nat. Genet. 2019 | 55374 |
| Autism Spectrum Disorder | ASD | Grove et al. Nat Genetics 2019 | 46350 |
| Bipolar Disorder | BPD | Stahl et al. (2019) | 51710 |
| Body Mass Index | BMI | Yengo et al. (2018) | 681275 |
| Educational Attainment | EDU | Lee et al. 2018 | 766345 |
| Epilepsy | EPI | Khalil, Nat. Comm. 2018 | 44889 |
| Extraversion | EXT | Van Den Berg et al. 2015 | 63030 |
| Height | HGT | Yengo et al. (2018) | 693529 |
| Intelligence | IQ | Savage et al. Nat. Genet. 2018 | 269867 |
| Major Depressive Disorder | MDD | Wray et al. Nat. Genet. 2018 | 480359 |
| Neuroticism | NEU | Nagel et al. Nat. Genet. 2018 | 380506 |
| Pain (chronic) | PAI | Johnston et al. PLoS Gen 2019 | 387649 |
| Parkinson's disease | PAR | Nalls et al., biorXiv 2019 | 482730 |
| Reaction Time | RT | Davies et al., Nat. Comm. 2018 | 282014 |
| Risky Behavior | RSK | Karlsson Linnér et al. Nat Gen. 2019 | 466571 |
| Schizophrenia | SCZ | Pardinas et al. Nat. Genet. 2018 | 105318 |
| Stroke (ischemic) | STR | Malik et al. Nat Gen 2018 | 440328 |
| Subjective Well-being | SWB | Okbay et al. 2016 | 298420 |

by maximum likelihood: we determined the exponential parameter that gave the highest probability for observing the full set of heritabilities estimated by LDSR across all phenotypes, taking into account the standard errors of these estimates (process documented in accompanying code). The posterior distribution of the estimate for each phenotype was then the exponential prior multiplied by the likelihood function, and the posterior estimates were computed as the expected value of the posterior distribution.

Empirical Bayes approaches introduce a bias in order to reduce unmodeled error. Since the aim of this paper is to document distinctions among phenotypes, and the bias of eBayes draws estimates for each phenotype toward the common mean of all phenotypes, the bias does not contribute to our results. We also tried a shrinkage strategy analogous to that used by the LASSO and found only very modest differences in results (not reported).

## 3. Results

The partitioned heritability estimates for the most significant categories and the enrichments (ratio of proportion of SNP-heritability to proportion of SNPs) for selected categories are shown in Fig 1; the raw LDSR estimates and their standard errors are presented in S1 Table. The classes contributing most to SNP-heritability were coding regions and transcription start sites (TSS; for most neurological disorders) and mammal-conserved sites (psychiatric and behavioral phenotypes). The most enriched classes (contributing much more than their proportion) were these three classes and also brain-specific promoters (mostly for psychiatric and behavioral).

The patterns of partitioned heritabilities seen in Fig 1 segregate with *a priori* classifications of the phenotypes, so we asked how the genetic architectures of the different traits relate to

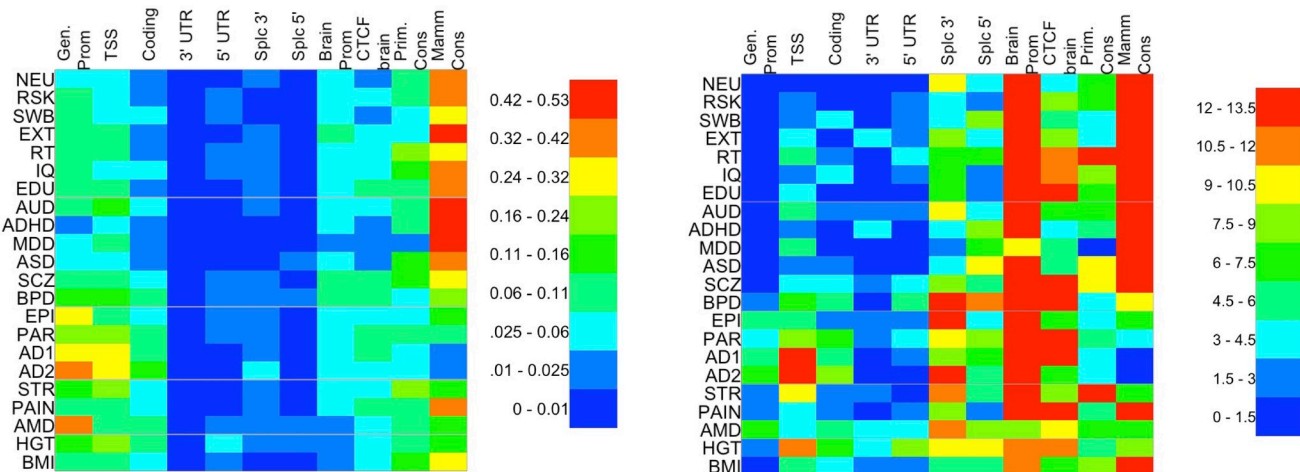

**Fig 1.** a) Empirical Bayes heritability estimates for the genomic classes studied here (in columns) for 20 traits and disorders (in rows). Color (legend at right) indicates estimated proportion of SNP-heritability. Estimates are (slightly) biased downward. b) Empirical Bayes enrichments of estimated SNP heritability attributed to various genomic classes by LDSR. Color indicates the enrichment (ratio of attributed heritability to proportion of SNPs) for each genomic category for each trait; key at right: blue: 0 (depletion); teal: little enrichment (1-2-fold); red: high (> 12-fold) enrichment. KEY: (for references see Table 2); AD1/2 Alzheimer's disease (see Table 2); ADHD: Attention Deficit Hyperactivity Disorder; ASD: Autism Spectrum Disorder; AMD: Age-related macular degeneration; AUD: Alcohol use disorder; BMI: Body mass index; BPD: Bipolar disorder; EDU: Educational Attainment; EPI: Epilepsy; EXT: Extraversion; HGT: Height; IQ: Intelligence quotient; MDD: Major depressive disorder; NEU: Neuroticism; PAI: Pain; PAR: Parkinson's disease; RSK: Risky Behavior; RT: Reaction Time; SCZ: Schizophrenia; STR: Stroke (all causes); SWB: Subjective well-being; Gen Prom: Generic promoter (from UCSC; non-overlapping with brain-specific promoters); TSS: Transcription start site; CTCF peaks: CTCF peaks from PsychENCODE brain samples.

each other. We represented the relations among partitioned heritability patterns of phenotypes (Fig 2) using isometric multi-dimensional scaling (MDS, using isoMDS in R3.3) We defined distance between phenotypes by the sum over categories of the absolute differences in estimated heritability. The heritability distribution patterns of the core psychiatric traits cluster together with behavioral traits at center-left, while neurological disorders are spread through the lower right.

The clustered arrangement of traits in Fig 2 suggests that the partition of heritability among classes might be robust enough to distinguish whether an unknown disorder was neurological or psychiatric. To test this rigorously, we fit a linear discriminant to the heritability partition vectors and performed leave-one-out cross-validation. The predicted out-of-sample classes were the same as actual classes in all cases, confirming that patterns of enrichment can help distinguish between neurological and psychiatric disorders. Fig 3 shows the loadings of the discriminant function. The contribution of mammal-conserved regions is the most discriminating measure, followed by contribution of coding regions (negative) and of primate-conserved regions. We were unable to find a robust linear discriminator based on genomic classes between behavioral traits and psychiatric disorders.

## 3.1 Results summary

We found that the majority of heritability for psychiatric disorders seems to be in putative regulatory enhancers (primate and mammal conserved) and in brain-specific promoters. The sum of estimated SNP-heritabilities over all categories was similar for most traits: between 80% and 90%. These results suggest that the categories used here, although comprising less than 13% of the common SNPs in the genome, account for most of the SNP-heritability of these disorders or traits. Furthermore at least half the SNP-heritability for psychiatric and behavioral phenotypes seems to lie in less than 3% of the genome.

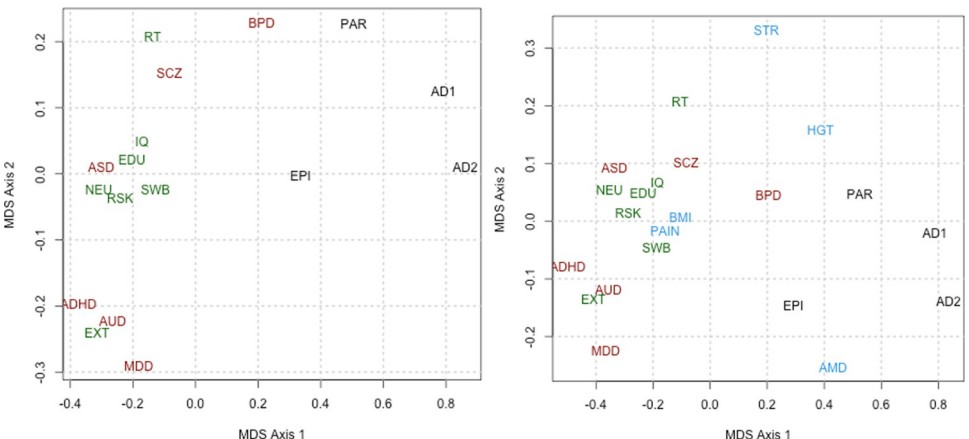

**Fig 2. Multi-dimensional scaling (Kruskal's isometric MDS implemented as isoMDS in R3.3) 2-D plots showing similarities of genetic architecture among different traits.** A) Representation of similarities among those traits thought to directly depend on nervous tissue. Note a clear separation of neurological and psychiatric disorders. The horizontal axis corresponds roughly to higher loadings on constitutive (coding, promoter, splicing) annotations toward the right and higher regulatory related loadings toward the left. The vertical axis corresponds to higher loadings on CTCF toward the bottom. B) Representation of similarities among all traits considered here. Note that stroke, and macular degeneration seem to define an axis orthogonal to the axis between behavioral disorders and traits and neurological disorders, and thus squash all the relations shown Fig 2A into one dimension. KEY: as for Fig 1.

Notably we have found that brain-specific promoters and conserved regions, otherwise unannotated, which were intended as proxies for putative inducible or cell-type specific enhancers, provide the majority of the SNP heritability for the major psychiatric disorders (schizophrenia, autism and bipolar disorder) as well as for behavioral traits, but not for neurological disorders.

## 4. Discussion

We sought to determine whether we could distinguish the functional genetic architectures of psychiatric disorders, behavioral traits and neurological disorders. We predicted that variation in regulatory sites would play a greater role in the etiology of psychiatric disorders and likely behavioral traits than in neurologic disorders, while the reverse pattern would be observed for coding sequence variation. Our results partially confirmed these expectations.

One of our new brain-specific categories–brain-specific promoters–contributed substantial heritability to psychiatric disorders and behavioral traits. The generic cross-tissue versions of this category used by [9] did not contribute substantially to psychiatric disorders, although the generic promoters did contribute to neurological disorders. We think it understandable that the generic promoters contribute more to neurological disorders since several of the neurological disorders involve the immune system, (e.g. cardiovascular events for stroke), and even those that affect neurons—Parkinson's Disease, Alzheimer's disease and epilepsy–seem to involve more basic molecular pathways than the psychiatric disorders, which seem to involve more subtle imbalances. Many genes have several promoters which may be active in different tissues. Use of different promoters will result in different 5'UTRs, which contain regulatory signals often related to trafficking the RNA to specific cell compartments, such as dendrites. The greater enrichment of brain-specific promoters validates our rationale for using regulatory sites derived specifically from brain chromatin data, with one significant exception: enhancers.

We identified brain promoters and expected to identify brain enhancers using data from published chromatin assays of human brain tissue [11]. Brain promoters selected from the annotations produced seemed useful, but LDSR did not find that enhancer annotations from

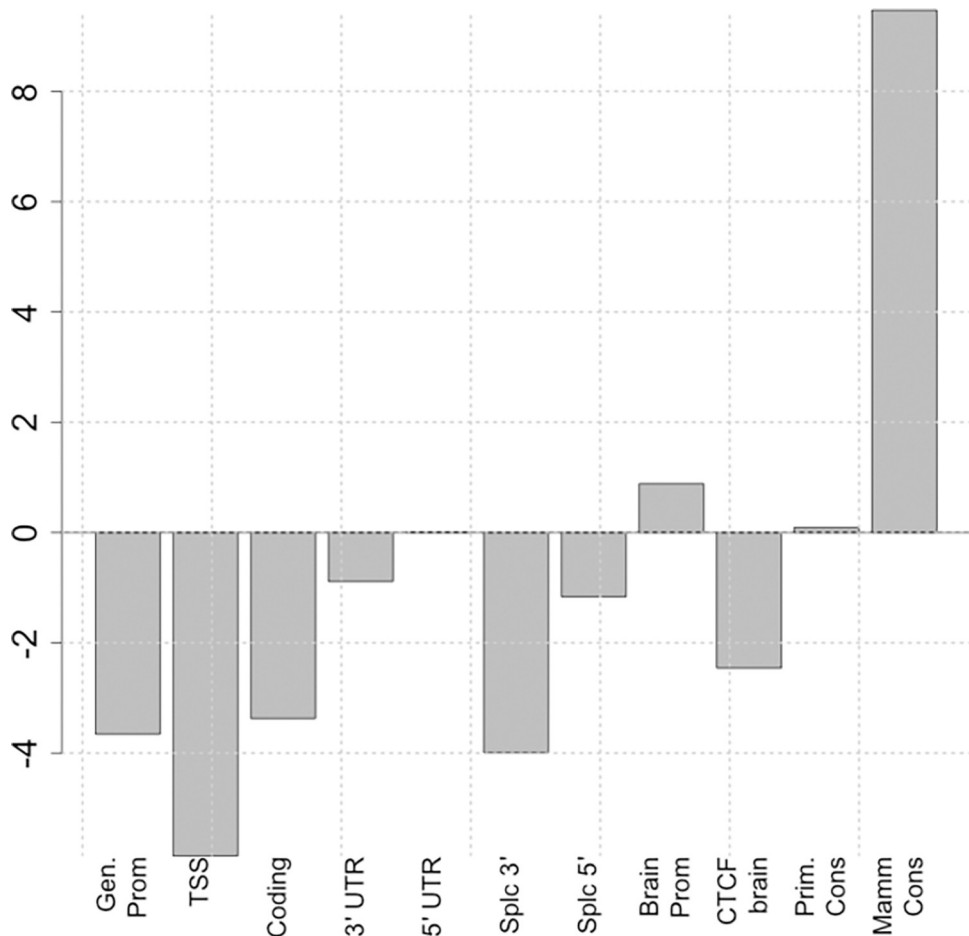

**Fig 3. Bar plot showing weights of the linear discriminant function separating SNP functional class enrichment profiles typical of psychiatric disorders and behavioral traits (positive enrichments) from those profiles typical of neurological disorders (negative enrichments).** Note heavy weighting on putative mammal and primate enhancers for psychiatric disorders, but on coding regions for neurological disorders. Note that because the proportions of different SNP classes vary by almost three orders of magnitude, the discriminant weights displayed here were determined for enrichment ratios (heritability for a class divided by proportion of SNPs in that class) rather than heritabilities.

these data sets explained a large fraction of SNP heritability. We suggest three main reasons for this. First, chromatin assays themselves are still crude. Second, currently available brain chromatin data is derived primarily from dissected tissue, aggregating across nuclei from all major cell types. Third, enhancers are not always active: many enhancers, especially those critical for learning are induced in only a small fraction of cells by specific signals in response to life experience, such as stress, and are 'on' for brief periods during which a burst of transcription is activated. We may expect future technical improvements to mitigate the first two issues, but the third will remain a challenge.

Enhancer annotations derived from the chromatin data currently available are thus likely to reflect predominantly constitutive enhancers in the most abundant cell types. Our success in finding enrichment signals in putative regulatory sites flagged by conservation, and our failure to find as much in chromatin data, suggests that inducible enhancers that are responsive to physiological signals or enhancers in minor cell types contribute to the genetics of psychiatric disorders more than other disorders. Our results thus strongly implicate inducible regulation

sites, or enhancers in minority cell types, in psychiatric disorders and behavior traits, but less so in neurologic disorders or biometric traits. It is thought that many enhancers are activated by signals of a departure from homeostasis, The second interpretation is consistent with evidence that interneurons [33] are implicated in psychiatric disorders. The first interpretation is consistent with evidence that physiological insults such as injury or infection or life stress [34] are implicated in psychiatric disorders more than in neurological disorders. We speculate that further refinement would identify many of the enhancers relevant for psychiatric disorders as responsive to specific life stresses, and that further research may bridge between the hitherto separate research programs of psychiatric genetics and environmental contributors to psychiatric disorders.

We were surprised to see that the functional genetic architecture of BMI seemed more similar to that of behavioral traits than to the standard biometric trait of height. However [35] found many SNPs relevant to BMI in or near genes expressed in the nervous system. The evidence from Fig 2 that bipolar disorder is intermediate between the psychiatric and classical neurologic and including epilepsy is noteworthy given the anticonvulsants are now standard maintenance treatments for bipolar disorder [36] and a number of the most prominent risk variants identified for this disorder are ion-channel genes [22].

Does the relationship between psychiatric disorders assessed from our functional genomic categories seen in Fig 2 map onto those obtained from common SNP variants formed into polygene scores? A definitive answer is not yet possible, but two lines of suggestive evidence can be derived from the magnitude of SNP-based genetic correlations that bear some resemblance to the distance between the disorders in Fig 2. First, using SCZ as an anchor point, SNP-based genetic correlations are high between SCZ and BPD (positioned relatively closely together Fig 2) and modest with MDD (which is much further apart) [1, 2]. Second, using AUD as an anchor, SNP genetic correlations are high with MDD and modest with SCZ and BPD [4] which corresponds to their relative placements in Fig 2.

These analyses raise the question of how much our pattern of findings might result from the pleiotropic effect of some proportion of the risk variants for these disorders. We can only say that this is not unlikely and further detailed work would be needed to quantify this effect.

The results presented here complement recent studies showing genes implicated by GWAS for neurological disorders concentrate in specific brain cell types, while genes implicated by GWAS for psychiatric disorders and behavioral traits are broadly enriched in telencephalic neurons [37].

## 5. Conclusion

In a novel use of LDSR, we have identified the genomic categories accounting for a majority of the SNP heritability for a number of major psychiatric disorders. We have also shown that the functional genetic architectures of many psychiatric disorders and behavioral traits are relatively similar to each other and less similar to the architectures of neurological diseases or to a control anthropometric trait like height. We have shown that distinctive genomic categories relevant to psychiatric disorders and behavioral traits are those related to dynamic gene regulation on short time scales. Our results hold promise for bridging genetics and well-established environmental and life-history risk factors for psychiatric disorders.

## Supporting information

**S1 Table. Numerical estimates from LDSC of heritability contributed by various annotations classes to the traits studied.**
(CSV)

## Acknowledgments

We thank Amanda Charbonneau Jorden Schossau, and Lindsay Guare for technical work with LDSR, and Bradley Verhulst for consultations about the GWAS used here.

## Author Contributions

**Conceptualization:** Mark A. Reimers, Kenneth S. Kendler.

**Data curation:** Mark A. Reimers.

**Formal analysis:** Mark A. Reimers.

**Funding acquisition:** Kenneth S. Kendler.

**Methodology:** Mark A. Reimers.

**Project administration:** Kenneth S. Kendler.

**Resources:** Kenneth S. Kendler.

**Supervision:** Mark A. Reimers, Kenneth S. Kendler.

**Visualization:** Mark A. Reimers.

**Writing – original draft:** Mark A. Reimers.

**Writing – review & editing:** Mark A. Reimers, Kenneth S. Kendler.

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
