## [Decision Letter · Decision Letter 0]

24 Feb 2021

PONE-D-21-03557

Functional classes of SNPs related to psychiatric disorders and behavioral traits contrast with those related to neurological disorders

PLOS ONE

Dear Dr. Reimers,

Thank you for submitting your manuscript to PLOS ONE. After careful consideration, we feel that it has merit but does not fully meet PLOS ONE’s publication criteria as it currently stands. Therefore, we invite you to submit a revised version of the manuscript that addresses the points raised during the review process.

We look forward to receiving your revised manuscript.

Kind regards,

Chunyu Liu

Academic Editor

PLOS ONE

Journal Requirements:

Reviewers' comments:

Reviewer's Responses to Questions

**Comments to the Author**

1. Is the manuscript technically sound, and do the data support the conclusions?

Reviewer #1: Partly

Reviewer #2: Yes

2. Has the statistical analysis been performed appropriately and rigorously? 

Reviewer #1: Yes

Reviewer #2: Yes

3. Have the authors made all data underlying the findings in their manuscript fully available?

Reviewer #1: Yes

Reviewer #2: Yes

4. Is the manuscript presented in an intelligible fashion and written in standard English?

Reviewer #1: Yes

Reviewer #2: Yes

5. Review Comments to the Author

Reviewer #1: The paper by Reimers et al. investigated the partitioned SNP-heritability of 20 brain phenotypes by the functional classes of genomic regions including several brain-specific genomic annotations like CTCF binding sites, brain-specific promoters, splicing sites. The authors compared the relative contributions of different functional classes of genomic regions on different brain disorders/traits and compared the enrichment performance between psychiatric disorders with neurological disorders. This study presented evidence to support that most GWAS SNP is functional in regulatory with less than 13% of the genome contribute to more than 80% of the SNP heritability and highlighted the CTCF binding sites, cell type-specific enhancers, and non-coding RNA in the psychiatric disorders.

Several major concerns need to be addressed:

1. Comparing psychiatric disorders and neurological diseases is one of the major works in this study, and the authors did a good job of comparing them from the perspective of different functional classes. However, there is no justification for the selection of brain phenotypes. In a Science paper by the brainstorm consortium (PMID: 29930110), they already compare the genetic correlation between psychiatric disorders and neurological disorders. It will be better to include all the traits in the paper to have a comparison or give justification in the selection of the phenotypes.

2. When comparing the different phenotypes, another important issue is pleiotropy, to what extend the similarity observed in this study is came from the pleiotropy is not discussed.

3. The authors highlighted that “the results provide a bridge between genetics and the well-known effects of life history and recent stressful experiences on the risk of psychiatric illness” with no direct evidence or related evaluation. It is better to change the description.

Minor concerns:

1. The figures are in low resolution.

2. The format of the tables is inappropriate.

3. The citation for LDSR is missing. The authors need to check the citation for methods.

4. The results for brain-specific CTCF are really interesting, more discussion for interpreting these results will be better.

Reviewer #2: The authors performed linkage-disequilibrium score regression among psychiatric disorders, behavioral traits and neurological disorders and found SNPs implicated in psychiatric disorders and behavioral traits were highly enriched in CTCF binding sites, in conserved regions likely to be enhancers, and in brain-specific promoters, regulatory sites likely to affect dynamic responses, while the SNPs relevant for neurological disorders were highly enriched in constitutive coding regions and splice regulatory sites. The study demonstrated the functional genetic architecture difference between psychiatric disorders and neurological disorders, which may bridge genetics and well-established environmental and life-history risk factors for psychiatric disorders. It is an interesting study. My suggestions regarding how this manuscript could be improved are stated below:

1. The results only showed heritability enrichment of the probable enhancers defined by authors, however, as showed in line 124-129 of manuscript, annotated enhancers (based on H3K27ac or ATAC chromatin peaks) explained more than 20% of SNP-heritability. How conserved of annotated enhancers, why the difference between defined enhancers and annotated enhancers and what advantages did defined enhancers have, please elaborate.

2. There were two Alzheimer's disease datasets analyzed separately, why not combined the two datasets by meta-analyze. Moreover, the 18 brain-related phenotypes described in line 166 counted Alzheimer's disease twice, and the “20 brain phenotypes” from title of figure 1 in line 212 was not precise.

3. As showed in line 175-181 of manuscript, LDSR method may yields negative heritability estimates when the trait had low SNP-heritability. However, the publicly available GWAS datasets sometimes cannot replicate the heritability estimates from the original study because some samples were excluded from released version, resulting underestimated heritability of the publicly available GWAS datasets. The authors should discuss the problem.

4. As shown in the Figure 1, psychiatric disorders and neurological disorders also showed some shared functional classes. The authors should note and discuss it.

5. I suggested the authors provided the same analysis in more non-psychiatric or non-neurological disorders as negative control.

6. The result is interesting, but how to explain it biologically?

7. All the Figures were not clear enough; the pixel was too low.

6. PLOS authors have the option to publish the peer review history of their article (what does this mean?). If published, this will include your full peer review and any attached files.

Reviewer #1: No

Reviewer #2: No

---

## [Author Response · Author response to Decision Letter 0]

16 Sep 2021

As reviewer #1 requested, we tried to include all of the traits in the BrainStorm consortium. We found that publicly reported results from several of the phenotypes were poorly suited for the LDSC tool, either because few SNPs were represented in the public databases (e.g. multiple sclerosis) or because of small case numbers (e.g. cerebral hemorrhage). Therefore we included in our analysis two additional traits - chronic pain and stroke - that were of sufficient quality to include in an LDSC calculation. Nevertheless we note that these additional traits are not diseases of neurons in the brain, even though they affect the brain. Therefore we reorganized our presentation of figure 2 into two new figures: a figure focused only on phenotypes likely to be due to functions of the nervous system, and one that includes our initial control traits and also the phenotypes that affect the central nervous system but do not seem to originate there.

Reviewer #2 pointed out that “the publicly available GWAS datasets sometimes cannot replicate the heritability estimates from the original study because some samples were excluded from the released version” and also suggested we combine the two GWAS for Alzheimer's. The reviewer raises a legitimate concern that the estimates, especially from a complex statistical method like LDSR, may depend sensitively on the data set or even the specific subset of data analyzed. 

In this regard, the estimates from the two independent AD GWAS are expected to be even further apart than estimates based on subsets of the same data; thus these two serve as a sanity check to suggest that the profiles are robust. Therefore we would prefer to report both separately, and have explained our rationale explicitly, although the reviewer is right to point out that they count as only one trait.

Reviewer # 2 also “suggested the authors provided the same analysis in more non-psychiatric or non-neurological disorders as negative control.” We have included some other brain-related but non-psychiatric or non-neurological disorders. Our interest is primarily in brain-related disorders, and we don’t expect our brain-specific annotations to be as useful in distinguishing the functional architecture of other phenotypes.

This reviewer also requests that we comment about the possible impact of a sharing of risk variants across the psychiatric and neurologic disorders. We add a few sentences on this in the revised discussion section. We also expand on the potential reason for the placement of bipolar disorder midway between the psychiatric and neurological disorders.

---

## [Decision Letter · Decision Letter 1]

2 Dec 2021

PONE-D-21-03557R1Functional classes of SNPs related to psychiatric disorders and behavioral traits contrast with those related to neurological disordersPLOS ONE

Dear Dr. Reimers,

Thank you for submitting your manuscript to PLOS ONE. After careful consideration, we feel that it has merit but does not fully meet PLOS ONE’s publication criteria as it currently stands. Therefore, we invite you to submit a revised version of the manuscript that addresses the points raised during the review process.

We look forward to receiving your revised manuscript.

Kind regards,

Chunyu Liu

Academic Editor

PLOS ONE

Journal Requirements:

Additional Editor Comments (if provided):

Besides the additional comments raised by one of the reviewers, I would recommend the authors to revise their conclusion in the abstract.

" We suggest that our results provide a bridge between genetics and the well-known effects of life history and recent

stressful experiences on risk of psychiatric illness."

I could not find how the findings are connected to either life history or recent stressful experience.

Is the enrichment of enhancer sufficient to relate to stress? I did not see a solid link there.

Reviewers' comments:

Reviewer's Responses to Questions

**Comments to the Author**

1. If the authors have adequately addressed your comments raised in a previous round of review and you feel that this manuscript is now acceptable for publication, you may indicate that here to bypass the “Comments to the Author” section, enter your conflict of interest statement in the “Confidential to Editor” section, and submit your "Accept" recommendation.

Reviewer #2: All comments have been addressed

Reviewer #3: All comments have been addressed

2. Is the manuscript technically sound, and do the data support the conclusions?

Reviewer #2: Yes

Reviewer #3: Yes

3. Has the statistical analysis been performed appropriately and rigorously? 

Reviewer #2: Yes

Reviewer #3: Yes

4. Have the authors made all data underlying the findings in their manuscript fully available?

Reviewer #2: Yes

Reviewer #3: Yes

5. Is the manuscript presented in an intelligible fashion and written in standard English?

Reviewer #2: Yes

Reviewer #3: Yes

6. Review Comments to the Author

Reviewer #2: The authors addressed all my questions, I have no further questions and it is ready to publication.

Reviewer #3: The authors addressed most of previous comments. However, several minor issues still need to be improved:

1. It is quite interesting that the authors found "The SNPs implicated in these psychiatric disorders and behavioral

traits were highly enriched in CTCF binding sites". Did any previous studies reported enrichment of risk SNPs associated with psychiatric disorders (or reuglatory SNPs) in CTCF binding sites ?

2. Did the authors account for the enrichment in CTCF binding sites might be due to the fact that there were many CTCF ChIP-seq datasets in ENCODE compared with other TFs ?

7. PLOS authors have the option to publish the peer review history of their article (what does this mean?). If published, this will include your full peer review and any attached files.

Reviewer #2: No

Reviewer #3: No

---

## [Author Response · Author response to Decision Letter 1]

6 Feb 2022

>Additional Editor Comments (if provided):

>Besides the additional comments raised by one of the reviewers, I would recommend the authors to revise their >conclusion in the abstract.

>" We suggest that our results provide a bridge between genetics and the well-known effects of life history and recent

>stressful experiences on risk of psychiatric illness."

I tried to clarify the discussion of the relation of our results to plasticity and stress but I deleted the sentence in the abstract under Additional Editor Comments. 

>Reviewer #3: The authors addressed most of previous comments. However, several minor issues still need to be >improved:

>1. It is quite interesting that the authors found "The SNPs implicated in these psychiatric disorders and behavioral

>traits were highly enriched in CTCF binding sites". Did any previous studies reported enrichment of risk SNPs >associated with psychiatric disorders (or reuglatory SNPs) in CTCF binding sites ?

>2. Did the authors account for the enrichment in CTCF binding sites might be due to the fact that there were many >CTCF ChIP-seq datasets in ENCODE compared with other TFs ?

The changes to correct the label switching (described in Response to Reviews), render these comments moot.

---

## [Editor Report · Decision Letter 2]

28 Feb 2022

PONE-D-21-03557R2Functional classes of SNPs related to psychiatric disorders and behavioral traits contrast with those related to neurological disordersPLOS ONE

Dear Dr. Reimers,

Thank you for submitting your manuscript to PLOS ONE. After careful consideration, we feel that it has merit but does not fully meet PLOS ONE’s publication criteria as it currently stands. Therefore, we invite you to submit a revised version of the manuscript that addresses the points raised during the review process. Please submit your revised manuscript by Apr 14 2022 11:59PM. If you will need more time than this to complete your revisions, please reply to this message or contact the journal office at plosone@plos.org. Please include the following items when submitting your revised manuscript:A rebuttal letter that responds to each point raised by the academic editor and reviewer(s). You should upload this letter as a separate file labeled 'Response to Reviewers'.A marked-up copy of your manuscript that highlights changes made to the original version. You should upload this as a separate file labeled 'Revised Manuscript with Track Changes'.An unmarked version of your revised paper without tracked changes. You should upload this as a separate file labeled 'Manuscript'.If applicable, we recommend that you deposit your laboratory protocols in protocols.io to enhance the reproducibility of your results. Protocols.io assigns your protocol its own identifier (DOI) so that it can be cited independently in the future. For instructions see: https://journals.plos.org/plosone/s/submission-guidelines#loc-laboratory-protocols. Additionally, PLOS ONE offers an option for publishing peer-reviewed Lab Protocol articles, which describe protocols hosted on protocols.io. Read more information on sharing protocols at https://plos.org/protocols?utm_medium=editorial-email&utm_source=authorletters&utm_campaign=protocols.

We look forward to receiving your revised manuscript.

Kind regards,

Chunyu Liu

Academic Editor

PLOS ONE

Journal Requirements:

Additional Editor Comments (if provided):

A few minor issues need to be addressed before we can accept this paper:

1. Several figures are not readable. Table 2 is chopped off. Please provide figures with good resolution.

2. Please clarify the differences between Gen promoter and Brain promoter in figure 3. I assume that "Gen promoter" is for all gene promoter, while Brain promoter refers to brain-specific promoter. A question here is about their possible overlap (?). Is Gen promoter mostly non-specific to brain? Please provide a bit more information. A little discussion about their distinct contributions to psychiatric and neurological diseases might be useful.

3. According to Fig 3, the CTCF weight is very small. How do we know that is a significant contributor? Moreover, since this CTCF is from PsychENCODE, it should be labeled as "brain CTCF," which could be very different from other tissue CTCF.

---

## [Author Response · Author response to Decision Letter 2]

20 Jul 2022

1. Several figures are not readable. Table 2 is chopped off. Please provide figures with good resolution.

I remade figures from code with higher resolution. 

I reformatted Table 2

2. Please clarify the differences between Gen promoter and Brain promoter in figure 3. I assume that "Gen promoter" is for all gene promoter, while Brain promoter refers to brain-specific promoter. A question here is about their possible overlap (?). Is Gen promoter mostly non-specific to brain? Please provide a bit more information. A little discussion about their distinct contributions to psychiatric and neurological diseases might be useful.

I defined 'Gen promoter' as 'generic promoter' in the Figure 1 legend, and noted no overlap with brain promoter by construction

3. According to Fig 3, the CTCF weight is very small. How do we know that is a significant contributor? Moreover, since this CTCF is from PsychENCODE, it should be labeled as "brain CTCF," which could be very different from other tissue CTCF.

The discriminant analysis does not readily return a measure of significance for each predictor, although this could be estimated indirectly by leave-one-out analysis. For dimensionless measures, as is the case here, the size of discriminant coefficient for a predictor is normally taken as a measure of its importance. Therefore, I included all bars in the new Figure 3, so readers may easily judge the relative importance.

---

## [Editor Report · Decision Letter 3]

16 Aug 2022

PONE-D-21-03557R3Functional classes of SNPs related to psychiatric disorders and behavioral traits contrast with those related to neurological disordersPLOS ONE

Dear Dr. Reimers,

Thank you for submitting your manuscript to PLOS ONE. After careful consideration, we feel that it has merit but does not fully meet PLOS ONE’s publication criteria as it currently stands. Therefore, we invite you to submit a revised version of the manuscript that addresses the points raised during the review process.

We look forward to receiving your revised manuscript.

Kind regards,

Chunyu Liu

Academic Editor

PLOS ONE

Journal Requirements:

Editor's comments

1. Please discuss the distinction of Gene promoter and brain promoter in Fig 3? Psychiatric disorders and neurological diseases are both brain-related. How could they be different? Is gene promoter all non-brain? Is it suggesting that neurological diseases have non-brain gene promoters affected? Sounds odd. 

2. The PsychENCODE brain CTCF is used in the analysis. Should we have a non-brain CTCF used for comparison too? The author had ENCODE CTCF before but dropped it in the revision. Why?

I am also surprised to see that brain CTCF was enriched in psychiatry in the previous version but was enriched in neurological diseases in the revised version (brain CTCF has a negative value, as shown in Fig 3). What’s changed to make such a switch?

3. There are several jumps from “brain promoter + conserved regions” to “enhancer” to “stress response in minority cell types.” Where is the evidence? Data? It seems to be more speculation.

---

## [Author Response · Author response to Decision Letter 3]

12 Sep 2022

Dear Editor

In response to your requests (numbered and copied below) about the last submission, we have done the following:

1. Please discuss the distinction of Gene promoter and brain promoter in Fig 3? Psychiatric disorders and neurological diseases are both brain-related. How could they be different? Is gene promoter all non-brain? Is it suggesting that neurological diseases have non-brain gene promoters affected? Sounds odd. 

We have distinguished more clearly generic promoter and brain promoter in the text

Since several of the neurological disorders heavily involve the immune system, (both microglia and cardiovascular events for stroke), think it is understandable that the generic promoters contribute more to those disorders. Even those neurologic disorders that are largely brain based on their pathology – Parkinson’s Disease, Alzheimer’s disease and epilepsy –seem to involve cellular pathologies quite different than the psychiatric disorders, which seem to involve more subtle imbalances. We have added a sentence to this effect in the Discussion. 

2. The PsychENCODE brain CTCF is used in the analysis. Should we have a non-brain CTCF used for comparison too? The author had ENCODE CTCF before but dropped it in the revision. Why?

I am also surprised to see that brain CTCF was enriched in psychiatry in the previous version but was enriched in neurological diseases in the revised version (brain CTCF has a negative value, as shown in Fig 3). What’s changed to make such a switch?

In the first two versions of this paper that we submitted, we had mistakenly switched the labels of columns for brain CTCF, which was corrected in the most recent two versions. We highlighted this change in the response letter for the third submission. That accounts for the switch in CTCF. The previous label was ‘CTCF peaks’, which has been re-labeled as brain CTCF in order to address a previous reviewer’s comments. These CTCF peaks came from PsychENCODE (different from ENCODE) as noted in Table 1. 

3. There are several jumps from “brain promoter + conserved regions” to “enhancer” to “stress response in minority cell types.” Where is the evidence? Data? It seems to be more speculation.

We argued explicitly in the text that a large fraction of conserved regions which are not explicitly identified as coding or promotor regions would be enhancers. It is thought that most enhancers are attachment points for specific DNA binding proteins, which are typically induced by particular cell signals. While there are many developmental enhancers, in the adult it is thought that most signals reflect a departure from homeostasis, hence a ‘stress’ in the generic sense. We think that various organismic stressors, such as infection, poverty, neglect or violence, will constitute departures from homeostasis. We have flagged this last conclusion as somewhat speculative in the Discussion.

Regards,

Mark Reimers and Kenneth Kendler

---

## [Editor Report · Decision Letter 4]

11 Oct 2022

PONE-D-21-03557R4Functional classes of SNPs related to psychiatric disorders and behavioral traits contrast with those related to neurological disordersPLOS ONE

Dear Dr. Reimers,

Thank you for submitting your manuscript to PLOS ONE. After careful consideration, we feel that it has merit but does not fully meet PLOS ONE’s publication criteria as it currently stands. Therefore, we invite you to submit a revised version of the manuscript that addresses the points raised during the review process.

We look forward to receiving your revised manuscript.

Kind regards,

Chunyu Liu

Academic Editor

PLOS ONE

Journal Requirements:

Additional Editor Comments :

In the introduction, the authors stated “We further expected long non-coding RNAs (lncRNAs) to contribute to psychiatric disorders because they were highly expressed in specific brain cell types and play critical roles during development.” So, this seems to be one of the two primary goals of this paper.

In the discussion, the authors had “On the theory of a developmental origin for psychiatric disorders, we expected to find substantial heritability for psychiatric syndromes in lncRNAs expressed during early development. Indeed, we found that all classes of noncoding RNAs appeared enriched across all phenotypes, consistent with the emerging idea that non-coding RNAs play a role in human brain diseases”.

But none of the results is related to lncRNA or early development. Where is the literature to claim lncRNA is early-development-specific? I have no clue how the authors came to this conclusion.

In the results, the authors showed: “We found that the three classes of non-coding RNAs tested (miRNAs, lncRNAs, rRNAs) appeared greatly enriched (medians 20-fold, 12-fold, and 20-fold, resp.) for contributions to SNP-heritability.” This is not for lncRNA only. I don’t see how it can be related to early development. There is no clear logic for me to follow. Moreover, there is not result shown for non-coding RNA enrichment, other than this description. If this is one of the two major goals, how could they have not even data to show?
---

## [Author Response · Author response to Decision Letter 4]

25 Nov 2022

EDITOR:

In the introduction, the authors stated “We further expected long non-coding RNAs (lncRNAs) to contribute to psychiatric disorders because they were highly expressed in specific brain cell types and play critical roles during development.” So, this seems to be one of the two primary goals of this paper.

In the discussion, the authors had “On the theory of a developmental origin for psychiatric disorders, we expected to find substantial heritability for psychiatric syndromes in lncRNAs expressed during early development. Indeed, we found that all classes of noncoding RNAs appeared enriched across all phenotypes, consistent with the emerging idea that non-coding RNAs play a role in human brain diseases”.

But none of the results is related to lncRNA or early development. Where is the literature to claim lncRNA is early-development-specific? I have no clue how the authors came to this conclusion.

In the results, the authors showed: “We found that the three classes of non-coding RNAs tested (miRNAs, lncRNAs, rRNAs) appeared greatly enriched (medians 20-fold, 12-fold, and 20-fold, resp.) for contributions to SNP-heritability.” This is not for lncRNA only. I don’t see how it can be related to early development. There is no clear logic for me to follow. Moreover, there is not result shown for non-coding RNA enrichment, other than this description. If this is one of the two major goals, how could they have not even data to show?

REPLY

Dear Editor,

Although several papers have argued for the importance of long non-coding RNA's in early brain development ( PMID: 27081004; PMID: 32554106; PMID: 31698782; PMID: 30728830), this claim is not at present well-established in the field. As you point out we do not actually present any data or results about long non-coding RNA’s, or indeed, any non-coding RNA's, because of wide error bars and hence insufficient statistical power. Therefore, we have deleted the discussion of enrichment and biological roles of non-coding RNA's, and kept only the technical mention of lack of statistical power, so that interested readers will know why they are not included.

---

## [Editor Report · Decision Letter 5]

14 Dec 2022

Functional classes of SNPs related to psychiatric disorders and behavioral traits contrast with those related to neurological disorders

PONE-D-21-03557R5

Dear Dr. Reimers,

We’re pleased to inform you that your manuscript has been judged scientifically suitable for publication and will be formally accepted for publication once it meets all outstanding technical requirements.

Kind regards,

Chunyu Liu

Academic Editor

PLOS ONE
---

## [Editor Report · Acceptance letter]

26 Dec 2022

PONE-D-21-03557R5 

Functional classes of SNPs related to psychiatric disorders and behavioral traits contrast with those related to neurological disorders 

Dear Dr. Reimers:

I'm pleased to inform you that your manuscript has been deemed suitable for publication in PLOS ONE. Congratulations! Your manuscript is now with our production department. 

Kind regards, 

on behalf of

Dr. Chunyu Liu 

Academic Editor

PLOS ONE